# Association between *Toxoplasma gondii* seropositivity and serointensity and brain volume in adults: A cross-sectional study

**Lance D. Erickson**[1], **Bruce L. Brown**[2], **Shawn D. Gale**[2,3]*, **Dawson W. Hedges**[2,3]

**1** Department of Sociology, Brigham Young University, Provo, Utah, United States of America, **2** Department of Psychology, Brigham Young University, Provo, Utah, United States of America, **3** The Neuroscience Center, Brigham Young University, Provo, Utah, United States of America

* shawn_gale@byu.edu

**Data Availability Statement:** Data underlying the results can be obtained through application to the UK Biobank http://www.ukbiobank.ac.uk. These data belong to the UK Biobank. This research has

## Abstract

The intracellular protozoal parasite *Toxoplasma gondii* has been associated with worsened cognitive function in animal models and in humans. Despite these associations, the mechanisms by which *Toxoplasma gondii* might affect cognitive function remain unknown, although *Toxoplasma gondii* does produce physiologically active intraneuronal cysts and appears to affect dopamine synthesis. Using data from the UK Biobank, we sought to determine whether *Toxoplasma gondii* is associated with decreased prefrontal, hippocampal, and thalamic gray-matter volumes and with decreased total gray-matter and total white-matter volumes in an adult community-based sample. The results from adjusted multivariable regression modelling showed no associations between *Toxoplasma gondii* and prefrontal, hippocampal, and thalamic brain gray-matter volumes. In contrast, natural-log transformed antibody levels against the *Toxoplasma gondii* p22 (b = -3960, 95-percent confidence interval, -6536 to -1383, p < .01) and sag1 (b = -4863, 95-percent confidence interval, −8301 to -1425, p < .01) antigens were associated with smaller total gray-matter volume, as was the mean of natural-log transformed p22 and sag1 titers (b = -6141, 95-percent confidence interval, -9886 to -2397, p < .01). There were no associations between any of the measures of *Toxoplasma gondii* and total white-matter volume. These findings suggest that *Toxoplasma gondii* might be associated with decreased total gray-matter in middle-aged and older middle-aged adults in a community-based sample from the United Kingdom.

## Introduction

Infecting approximately one-third of the world's population [1], the neurotropic intracellular apicomplexan protozoal parasite *Toxoplasma gondii* can remain in the brain for the life of the host [2]. The definitive hosts for *Toxoplasma gondii* are members of the cat family, which release oocysts into the environment, from where they can infect humans via contact with cat feces, ingestion of undercooked meat containing *Toxoplasma gondii* oocysts, or congenital transmission [3].

Some research fails to identify associations of *Toxoplasma gondii* seropositivity and serointensity with cognitive impairment [4–6], and one study even found an association between

been conducted using the UK Biobank Resource under Application Number 41535.

**Funding:** The authors received no specific funding for this work.

**Competing interests:** The authors have declared that no competing interests exist.

*Toxoplasma gondii* and better action control and smaller P300 event-related potential, which could indicate better cognitive function in certain cognitive tasks [7]. However, several studies have found associations between *Toxoplasma gondii* and impaired cognitive function [8–14]. One recent study using data from the UK Biobank found that *Toxoplasma gondii* was associated with worse executive function in adults, even after adjusting for several potentially confounding variables [15]. Furthermore, meta-analyses have identified associations of *Toxoplasma gondii* with dementia [16] and epilepsy [17]. Finally, in a sample of older adults, participants seropositive for *Toxoplasma gondii* had worse working memory, smaller P3b amplitude with event-related potentials, and reduced evoked frequency in the theta range compared to seronegative participants [18]. Together, these findings suggest that *Toxoplasma gondii* might be adversely associated with brain function.

Despite mounting evidence suggesting that *Toxoplasma gondii* adversely affects brain function in humans, the mechanisms underlying these associations remain unknown, although findings suggest several possible indirect mechanisms. Among potential indirect mechanisms, *Toxoplasma gondii* can increase the permeability of the gastrointestinal-blood barrier [2], which could enable increased toxin entry into the blood and from there into the brain. In addition, *Toxoplasma gondii* forms intraneuronal cysts that appear to be metabolically active [19], possibly affecting brain dopamine, glutamate, gamma amino butyric acid, and serotonin [20], neurotransmitters that could affect cognitive and brain function. *Toxoplasma gondii* also appears to affect gene expression [21]. Therefore, there are several possible mechanisms through which *Toxoplasma gondii* could affect brain function, potential mechanisms that could operate simultaneously.

Given the associations between *Toxoplasma gondii* and evidence of adverse effects on brain function and potential mechanisms by which it could affect neuronal function, *Toxoplasma gondii* could be related to adverse brain and cognitive functioning by affecting brain volume. Brain volume is weakly but consistently associated with cognitive function including intelligence [22]. In a community-based sample, brain volume in later life was concurrently associated with cognitive function [23]. To date, however, only a few studies have investigated associations between *Toxoplasma gondii* and brain volume. In a murine model, mice infected with *Toxoplasma gondii* for at least a year after initial infection in early adulthood had increased ventricular volume and elevated markers of neuron death [24]. In humans, a voxel-based morphometry study of 44 participants with schizophrenia and 56 healthy controls [25] found an association between *Toxoplasma gondii* and decreased gray-matter density among the participants with schizophrenia but not in the control participants.

Based on findings of possible adverse brain function including cognition associated with *Toxoplasma gondii* and preliminary evidence that it might alter brain volume, we sought to investigate further whether *Toxoplasma gondii* might be associated with brain volume using a larger sample than that used in the previously reported study [25], hypothesizing that *Toxoplasma gondii* would be associated with reduced brain volume. To test this hypothesis, we used a community-based dataset from the UK Biobank of middle and late middle-aged adults that contains data describing infection with *Toxoplasma gondii*, volumetric magnetic resonance imaging data, and demographic data (http://www.ukbiobank.ac.uk).

## Methods

### Study sample

The participants in this study are from the UK Biobank, a community-based sample of about 500,000 adults enrolled at ages 40 to 69 years between 2006 and 2010 and sampled from population-based registries (http://www.ukbiobank.ac.uk) [26]. In this database, there were 21,402

participants who had magnetic resonance brain imaging data (UK Biobank Brain Imaging Documentation, http://www.ukbiobank.ac.uk) and 9,431 participants who had serological data for exposure to *Toxoplasma gondii*. Only 434 participants, however, were eligible for our analyses in that they had both magnetic resonance imaging data and serological data for *Toxoplasma gondii*. Our final analytic sample was 385 due to missing data on our preidentified control variables. Although the UK Biobank sample is not representative of the UK population, the dataset can still be used to establish valid exposure-outcome associations (http://www.ukbiobank.ac.uk/wp-content/uploads/2017/03/access-matters-representativeness-1.pdf). The UK Biobank received ethical approval to collect demographic and medical data (reference 11/NW/0382), and all participants provided informed consent (http://biobank.ctsu.ox.ac.uk/crystal/field/cgi?id=200). We received regulatory approval from the UK Biobank to use data from the UK Biobank for our research (http://biobank.ctsu.ox.ac.uk/crystal/field.cgi?id=200).

## Brain volumes

We used pre-processed magnetic-resonance brain imaging data that the UK Biobank had obtained between 30 April 2014 and 19 March 2019 acquired using a 3-Tesla, 32-channel coil Siemens Skyra scanner (Siemens Medical Solutions, Germany; http://biobank.ctsu.ox.ac.uk/crystal/label.cgi?id=100003) [27, 28]. In our analyses, we used brain-imaging data for gray matter in mm$^3$ for the left and right frontal pole, the left and right superior frontal gyrus, the left and right medial frontal cortex, the left and right orbital frontal cortex, and the left and right frontal operculum, the left and right hippocampus, and the left and right thalamus. We also used brain-imaging data for total brain gray matter and white matter in mm$^3$.

## *Toxoplasma gondii*

The *Toxoplasma gondii*-related antibodies present in the UK Biobank were p22 and sag1 antibodies, measured in units of median florescence intensity [29]. The UK Biobank guidelines indicate a person is seropositive for *Toxoplasma gondii* if either the sag1 titer is greater than 160 or the p22 titer is greater than 100 (http://biobank.ndph.ox.ac.uk/showcase/field.cgi?tk=BVm2NA8JqLsnU55EBZF5XntyJ9AnSZMB1000738&id=23062). In statistical models, we included an indicator of *Toxoplasma gondii* seropositivity that was equal to 1 if participants were seropositive based on these criteria or 0 if they were negative. To capture any linear relationship between these antibodies and brain size, we independently examined the natural log of p22 and sag1 titers. We also examined the mean of standardized versions of natural-logged p22 and sag1 titers.

## Covariates

To control for variables that potentially could confound associations between *Toxoplasma gondii* and brain volume, we adjusted our statistical models for several variables that have been associated with cognitive function [30–32] or that plausibly could affect an association with brain volume, including age in years, sex (female, male), race/ethnicity (White, non-White), and educational attainment (having obtained a college degree compared to having obtained less than a college degree). Annual household income was originally recorded in five categories: (less than £18,000, £18,000 to £30,999, £31,000 to £51,999, £52,000 to £99,999, and greater than £100,000). We recoded responses to the middle value of each category, represented in £10,000 units (i.e. £18,000 to £30,999 became 2.45). We also included self-rated health (four-point scale ranging from poor to excellent), body-mass index (weight in kilograms/height in meters squared), smoking history (non-smoker, past, current), and frequency of alcohol use (six categories ranging from never to almost daily or daily). Statistical models of

the five regions in the prefrontal cortex, hippocampus, and thalamus included an additional control of total brain volume, which was the volume of gray and white matter normalized for head size.

## Statistical analysis

We estimated a series of models for each of the left and right brain regions (i.e., five prefrontal regions, the hippocampus, and the thalamus), with each model including a different focal independent variable (i.e. *Toxoplasma gondii* seropositivity, natural-log transformed sag1 antibodies, natural-log transformed p22 antibodies, and the mean of the standardized natural-log transformed sag1 and p22 antibodies). Each model included all covariates as well as total brain size. Subsequently, using these models as a baseline, we estimated additional models, each with a separate interaction of one measure of *Toxoplasma gondii* with age, sex, educational attainment, or race/ethnicity.

In total, this represents 56 statistical tests of interest in models without interactions and an additional 224 statistical tests for the interactions (i.e., 56 for each of four *Toxoplasma gondii* by control interactions). Consequently, to protect against alpha inflation due to multiple testing, we estimated a series of multivariate tests in which we tested a single independent variable and multiple dependent variables [33]. In doing this, we submitted all models where *Toxoplasma gondii* seropositivity was the focal independent variable in the absence of interactions (see Table 2) to a multivariate test that encompassed the joint covariance between the independent variable (i.e., *Toxoplasma gondii*) and dependent variables (i.e., the combination of left and right brain region volumes). We did this using Stata's *suest* command [34], which produces a single parameter vector for all of the models that takes into account the joint covariance of the dependent variables and therefore allows a test whose null hypothesis is that the joint relationship between the predictor and the dependent variables is zero. If the multivariate test was not significant, we did not consider any significant individual test to be significant. Conversely, if the multivariate test was significant, we concluded that any significant individual estimate was indeed valid.

In a second set of analyses, we estimated a series of models of total gray-matter and total white-matter volume for the seropositivity, p22, sag1, and mean of p22 and sag1 variables. We also estimated subsequent models, each with an interaction term of each of the measures of *Toxoplasma gondii* with age, sex, educational attainment, and income. Although there were far fewer models in the second set of analyses compared to the first, we again estimated multivariate tests to protect against type-1 errors due to multiple testing [33].

We used Stata 16.1 (StataCorp, Stata Statistical Software, Release 16. College Station, Texas) for all analyses.

## Results

The average age of the sample was 62.02 years, and 55 percent of the sample were women. Fifty-three percent of the sample had attained a college degree, and 96 percent were White. Twenty-six percent of the sample were seropositive for *Toxoplasma* gondii (Table 1). Table 1 also shows means, standard deviations (for continuous variables), and minimum and maximum values for all demographic, brain imaging, and *Toxoplasma gondii* variables that we included in the statistical models. By visual inspection, the prefrontal, hippocampal, and thalamic brain regions and the total gray-matter and total-white matter volumes were normally distributed.

There were no significant associations between any of the measures of *Toxoplasma gondii* serology and prefrontal, hippocampal, or thalamic volumes (Table 2). Because of the uniform

**Table 1. Descriptive statistics of study variables.**

| | Mean | SD | Minimum | Maximum |
|---|---|---|---|---|
| Prefrontal brain volume (mm$^3$) | | | | |
| Pole (left) | 23449.58 | 2861.91 | 16896.00 | 33728.60 |
| Pole (right) | 26515.12 | 3158.09 | 18330.40 | 36951.10 |
| Superior gyrus (left) | 11118.54 | 1689.33 | 6940.39 | 17040.00 |
| Superior gyrus (right) | 9700.46 | 1599.00 | 5946.68 | 15202.90 |
| Medial cortex (left) | 1914.28 | 334.54 | 1137.32 | 2934.77 |
| Medial cortex (right) | 1923.88 | 341.08 | 1121.24 | 3296.26 |
| Orbital cortex (left) | 6678.90 | 882.68 | 4417.84 | 10268.60 |
| Orbital cortex (right) | 6026.93 | 776.97 | 4045.75 | 8493.09 |
| Operculum cortex (left) | 1507.08 | 256.05 | 805.67 | 2357.56 |
| Operculum cortex (right) | 1351.52 | 250.79 | 630.69 | 2145.18 |
| Hippocampus volume (mm$^3$) | | | | |
| Hippocampus (left) | 3800.28 | 500.21 | 1811.00 | 5287.00 |
| Hippocampus (right) | 3906.89 | 502.13 | 2332.00 | 5163.00 |
| Thalamus volume (mm$^3$) | | | | |
| Thalamus (left) | 7780.30 | 750.59 | 5499.00 | 9896.00 |
| Thalamus (right) | 7592.70 | 729.83 | 5254.00 | 9566.00 |
| Total brain volume (mm$^3$) | 1508803.82 | 73428.04 | 1299110.00 | 1700560.00 |
| Gray matter | 800991.22 | 47779.65 | 672942.00 | 919510.00 |
| White matter | 707812.52 | 40316.62 | 598262.00 | 830308.00 |
| *Toxoplasma gondii* | | | | |
| Seropositive | .26 | | .00 | 1.00 |
| ln(p22) | 3.34 | 1.37 | .00 | 8.88 |
| ln(sag1) | 4.39 | 1.04 | .00 | 6.70 |
| Mean of ln(p22) & ln(sag1) | -.06 | .94 | -3.59 | 2.28 |
| Age | 62.02 | 7.42 | 47.00 | 78.00 |
| Female | .55 | | .00 | 1.00 |
| White | .96 | | .00 | 1.00 |
| College degree | .53 | | .00 | 1.00 |
| Income (in 10,000 £) | 4.43 | | .90 | 12.50 |
| Overall health | 2.98 | .67 | 1.00 | 4.00 |
| Body-mass index | 26.26 | 4.20 | 16.68 | 46.88 |
| Smoking status | | | | |
| Non-smoker | .63 | | .00 | 1.00 |
| Past | .34 | | .00 | 1.00 |
| Current | .03 | | .00 | 1.00 |
| Drinking frequency | | | | |
| Daily or almost daily | .18 | | .00 | 1.00 |
| 3–4 times/week | .25 | | .00 | 1.00 |
| Once or twice/week | .27 | | .00 | 1.00 |
| 1–3 times/month | .12 | | .00 | 1.00 |
| Special occasions | .13 | | .00 | 1.00 |
| Never | .04 | | .00 | 1.00 |

Note: N = 385. Source: *UK Biobank*.

**Table 2. Relationship between *Toxoplasma gondii* and prefrontal brain volume (mm3): Unstandardized coefficients and 95% confidence intervals from linear regression[a].**

| | Frontal | | | | | Hippocampus | Thalamus | Multivariate *p* |
|---|---|---|---|---|---|---|---|---|
| | Pole | Superior gyrus | Medial cortex | Orbital cortex | Operculum cortex | | | |
| Seropositive | | | | | | | | .063 |
| Left | -403.87 | -103.53 | -34.68 | -10.40 | -10.31 | -103.62 | 75.98 | |
| | -974.50,166.76 | -465.60,258.54 | -108.69,39.34 | -191.75,170.94 | -65.01,44.40 | -211.35,4.11 | -58.18,210.15 | |
| Right | -339.43 | -312.21 | -24.90 | 63.28 | -33.86 | -69.67 | 52.47 | |
| | -961.72,282.85 | -664.64,40.22 | -100.23,50.43 | -90.92,217.48 | -88.12,20.41 | -178.78,39.45 | -77.82,182.76 | |
| ln(p22) | | | | | | | | .078 |
| Left | -82.80 | -11.56 | -.63 | -18.43 | 2.55 | -27.44 | 33.24 | |
| | -268.26,102.66 | -129.10,105.97 | -24.67,23.42 | -77.25,40.38 | -15.20,20.30 | -62.46,7.57 | -10.24,76.71 | |
| Right | -90.88 | -80.15 | 3.14 | 21.50 | -7.25 | -8.38 | 24.34 | |
| | -292.91,111.15 | -194.69,34.39 | -21.31,27.60 | -28.53,71.54 | -24.88,10.38 | -43.85,27.09 | -17.90,66.58 | |
| ln(sag1) | | | | | | | | .163 |
| Left | -151.30 | 21.44 | -6.94 | -26.92 | .72 | -21.94 | 33.88 | |
| | -396.65,94.05 | -134.19,177.07 | -38.77,24.89 | -104.79,50.96 | -22.79,24.23 | -68.40,24.52 | -23.77,91.52 | |
| Right | -54.08 | -60.81 | 14.28 | 4.74 | 5.48 | -27.42 | 23.98 | |
| | -321.83,213.68 | -212.75,91.12 | -18.08,46.63 | -61.57,71.06 | -17.88,28.84 | -74.32,19.49 | -32.00,79.96 | |
| Mean of ln(p22) & ln(sag1) | | | | | | | | .124 |
| Left | -159.89 | 4.46 | -4.86 | -31.34 | 2.44 | -35.24 | 47.28 | |
| | -429.14,109.36 | -166.32,175.24 | -39.79,30.08 | -116.78,54.11 | -23.35,28.24 | -86.15,15.67 | -15.90,110.45 | |
| Right | -105.00 | -100.87 | 11.44 | 19.78 | -2.21 | -23.80 | 34.10 | |
| | -398.66,188.65 | -267.39,65.66 | -24.08,46.96 | -52.96,92.51 | -27.85,23.42 | -75.30,27.70 | -27.28,95.49 | |

Note:

[a] Each coefficient and associated confidence interval is from a different model that includes controls for age, sex, race, educational attainment, household income, self-rated health, body-mass index, smoking status, drinking frequency.

^b Multivariate tests test the multivariate significance of each pollutant on all of the brain volume outcomes. N = 385. Source: UK Biobank. * $p < .05$. ** $p < .01$. *** $p < .001$.

absence of statistically significant relationships in these analyses, we considered whether the analyses were underpowered because of the small sample size. Accordingly, we estimated post-hoc power analyses for each model to determine the sample size needed to identify a statistically significant result. We anticipated a biologically and clinically meaningful effect size to be close to the size of brain volume change typical for a one-year increase in aging [35]. The power analysis compared the model $R^2$ of the model reported in Table 2 (i.e., a "full" model) with a "reduced" model that excluded age. The results of these comparisons are the sample sizes that would be needed to identify a statistically significant coefficient for age with 80-percent power and an alpha of .05.

The power analyses (Table 3) showed that our sample of 385 individuals did contain enough power to identify a significant effect with 80-percent power and an alpha of .05 for *Toxoplasma gondii* if the effect were as large as the age effect for the left frontal pole, the left and right orbital frontal cortex, the left frontal operculum, the left hippocampus, and the left and right thalamus. However, the analyses for the right frontal pole, the left and right superior frontal gyrus, the left and right medial frontal cortex, the right frontal operculum, and the right hippocampus were underpowered. The sample size estimates for the superior frontal gyrus were substantial. This suggests that age was not a useful variable to assess the power of our statistical model of superior frontal gyrus volumes.

**Table 3. Power analyses: Required sample size for age effect on brain size to be significant.**

| | Frontal | | | | | Hippocampus | Thalamus |
| | Pole | Superior gyrus | Medial cortex | Orbital cortex | Operculum cortex | | |
|---|---|---|---|---|---|---|---|
| Seropositive | | | | | | | |
| Left | 244 | 32315 | 501 | 170 | 275 | 386 | 141 |
| Right | 628 | 5769494 | 495 | 120 | 840 | 2626 | 169 |
| ln(p22) | | | | | | | |
| Left | 231 | 26567 | 495 | 166 | 280 | 341 | 152 |
| Right | 574 | 126582 | 503 | 126 | 759 | 2315 | 181 |
| ln(sag1) | | | | | | | |
| Left | 240 | 29333 | 493 | 170 | 273 | 377 | 142 |
| Right | 615 | 9458555 | 486 | 121 | 815 | 2499 | 171 |
| Mean of ln(p22) & ln(sag1) | | | | | | | |
| Left | 232 | 30084 | 488 | 167 | 276 | 358 | 147 |
| Right | 596 | 464343 | 502 | 122 | 811 | 2294 | 175 |

Note: N = 385. Source: UK Biobank.

Table 4 presents the p-value of the multivariate tests of interaction terms in models examining whether age, sex, education, or income, moderate the relationship of *Toxoplasma gondii* and the five regions in the prefrontal cortex, the hippocampus, and the thalamus. The results indicate significant multivariate interactions between the four *Toxoplasma gondii* variables and age as well as between *Toxoplasma gondii* seropositivity and education.

The model-specific results presented in S1 through S4 Tables in S1 File indicate that none of the univariate interactions between *Toxoplasma gondii* and age was significant. Of the 224 interaction terms from adjusted models represented in the S1 File, only the interactions between *Toxoplasma gondii* seropositivity and educational attainment associated with gray-matter volume in the left superior frontal gyrus and in the left thalamus were statistically significant (S3 Table in S1 File). These two significant interactions represent less than one percent of the interaction models estimated. Therefore, although the associated multivariate test was significant (p = .020), the preponderance of evidence suggests that the relationship between *Toxoplasma gondii* and these brain volume regions does not vary by age, sex, education, or income.

**Table 4. Multivariate p-values from adjusted models[a] of the interactions of *Toxoplasma gondii* of the five left and right prefrontal regions, the left and right hippocampus, and the left and right thalamus with age, sex, education, and income.**

| | *T. gondii* seropositive | ln(p22) | ln(sag1) | Mean of ln(p22) & ln(sag1) |
|---|---|---|---|---|
| Age x *T. gondii* | .016 | .002 | .049 | .005 |
| Female x *T. gondii* | .546 | .441 | .119 | .605 |
| College degree x *T. gondii* | .020 | .189 | .132 | .086 |
| Income x *T. gondii* | .167 | .242 | .346 | .220 |

Note:

[a] Each p-value represents a multivariate test which is a test of the null hypothesis considered within the joint covariance of the dependent variables (i.e., the left and right side of the five prefrontal regions, the hippocampus, and thalamus) and the respective interaction between one of the *T. gondii* variables and a predictor (e.g., Age x *T. gondii* seropositive).

Results of the corresponding univariate statistical tests of these sixteen multivariate tests are presented in S1 through S4 Tables in S1 File. N = 385. Source: *UK Biobank*.

**Table 5. Relationship between *Toxoplasma gondii* and total brain volume (mm3): Unstandardized coefficients and 95% confidence intervals from linear regression[a].**

| | Gray Matter | | White Matter | | Multivariate *p* |
|---|---|---|---|---|---|
| | b | 95%CI | b | 95%CI | |
| Seropositive | -6074 | -14155,2007 | 583 | -8217,9382 | .295 |
| ln(p22) | -3960** | -6536,-1383 | -1828 | -4654,998 | .008 |
| ln(sag1) | -4863** | -8301,-1425 | 92 | -3679,3864 | .014 |
| Mean of ln(p22) & ln(sag1) | -6141** | -9886,-2397 | -1375 | -5495,2746 | .004 |

Note:

[a] Each coefficient and associated confidence interval is from a different model that includes controls for age, sex, race, education, income, self-rated health, body-mass index, smoking status, drinking frequency.

N = 385. Source: *UK Biobank*.

* $p < .05$.

** $p < .01$.

*** $p < .001$.

In contrast to the findings for associations between the measures for *Toxoplasma gondii* infection and prefrontal, hippocampal, and thalamic gray-matter volumes, Table 5 shows that the natural-log transformed p22 antibody titers (b = -3960, 95-percent confidence interval, -6536 to -1383, $p < .01$) and the natural-log transformed sag1 antibody titers (b = -4863, 95-percent confidence interval, −8301 to -1425, $p < .01$) were associated with smaller total gray-matter volume, as was the mean of natural-log transformed p22 and sag1 titers (b = -6141, 95-percent confidence interval, -9886 to -2397, $p < .01$). There were no significant associations between any of the measures of *Toxoplasma gondii* infection and total white-matter volume.

Table 6 presents results of the multivariate tests of the interaction of *Toxoplasma gondii* with age, sex, education, and income with total gray-matter and total white-matter volumes as the dependent variables (individual models are presented in S5 Table in S1 File). Interactions between *Toxoplasma gondii* seropositivity and sex and between *Toxoplasma gondii* (all four measures) and income were associated with total gray-matter and total white-matter volumes. Out of 32 interaction models, seven were statistically significant, each of which withstood multivariable testing for multiple comparisons.

**Table 6. Multivariate p-values from adjusted models[a] of the interactions of *Toxoplasma gondii* of total brain gray-matter volume and total brain white-matter volume with age, sex, education, and income.**

| | *T. gondii* seropositive | ln(p22) | ln(sag1) | Mean of ln(p22) & ln(sag1) |
|---|---|---|---|---|
| Age x *T. gondii* | .203 | .419 | .089 | .154 |
| Female x *T. gondii* | .022 | .255 | .877 | .663 |
| College degree x *T. gondii* | .232 | .137 | .503 | .285 |
| Income x *T. gondii* | .010 | .007 | .006 | .003 |

Note:

[a] Each p-value represents a multivariate test which is a test of the null hypothesis considered within the joint covariance of the dependent variables (i.e., total brain gray-matter and white-matter volumes) and the respective interaction between one of the *T. gondii* variables and a predictor (e.g., Age x *T. gondii* seropositive).

Results of the corresponding univariate statistical tests for total brain gray matter and white matter that are represented in these multivariate tests are presented in S5 Table in S1 File. N = 385. Source: *UK Biobank*.

## Discussion

The main finding from this community-based sample of middle-aged and late-middle-aged adults in the United Kingdom is an association between *Toxoplasma gondii* serointensity and smaller total gray-matter volume but no associations with total white-matter volume. Further, there were no associations between *Toxoplasma gondii* seropositivity and serointensity and gray-matter volume in five regions of the prefrontal cortex we evaluated, the hippocampus, and the thalamus.

The finding of a negative association between *Toxoplasma gondii* and total gray-matter volume suggests that changes (i.e., reductions) in gray-matter volume could be a mechanism of the relationship between *Toxoplasma gondii* and previously reported evidence of abnormal brain function, including worse cognitive functioning. While the smaller total gray-matter volume associated with *Toxoplasma gondii* serointensity could be a mechanism by which *Toxoplasma gondii* influences brain and cognitive function, we did not find that gray-matter volume in five different prefrontal regions, the hippocampus, or the thalamus was associated with either *Toxoplasma gondii* seropositivity or serointensity. Therefore, to the extent that reductions in gray-matter volume could be a mechanism of the relationship between *Toxoplasma gondii* and brain and cognitive functioning based on the association between *Toxoplasma gondii* and total gray-matter volume, the locale of the effect may be in brain regions we could not explore here, or a diffuse loss of gray matter might be associated with the decrease in brain and cognitive function associated with *Toxoplasma gondii* that some previous studies [8–14] have found. Further, even if reductions in brain volume is a mechanism of the relationship between *Toxoplasma gondii* and brain and cognitive functioning, other mechanisms could operate as well.

Few previous studies have investigated associations between *Toxoplasma gondii* serointensity and seropositivity and brain volume. Our results, however, are consistent with a previous study that found increased ventricular volume in mice infected with *Toxoplasma gondii* [24] and somewhat consistent with those of one previous study that found an association between *Toxoplasma gondii* and gray-matter density in schizophrenia, although they differ from other findings from the latter study that showed no association between gray-matter density and toxoplasmosis in healthy controls [25]. Possible reasons for these differences could be due to different samples, different control variables, and differences in statistical power, as the previous study included only 56 healthy controls of whom only 13 were infected with *Toxoplasma gondii*, [25] and might have been underpowered to detect associations between *Toxoplasma gondii* and brain volume in the healthy control group.

We found no meaningful interactions between *Toxoplasma gondii* and age, sex, educational attainment, and income associated with gray-matter volume in the five regions of the prefrontal cortex, the hippocampus, and the thalamus. Because only 2 out of 224 interaction models (less than one percent) were significant, the overwhelming pattern in this set of results suggests that there were no statistically significant interactions. In contrast, seven out of 32 (22 percent) interactions between *Toxoplasma gondii* and sex and between *Toxoplasma gondii* and income were significantly associated with total gray-matter and total white-matter volumes even after multivariate correction for multiple testing. There were significant interactions between *Toxoplasma gondii* seropositivity and sex and both total gray-matter and total white-matter volumes, between *Toxoplasma gondii* seropositivity and income and both total gray-matter and total white-matter volumes, between the natural-log transformed p22 titer and income and total white-matter volume, between the natural-log transformed sag1 titer and income and total white-matter volume, and between the mean of the natural-log transformed p22 and sag1 titers and income and total white-matter volume. Together, the significant interaction models

associated with total gray-matter and total white-matter volumes suggest that sex and income might moderate associations between *Toxoplasma gondii* serointensity and total gray-matter and total white-matter volumes.

Based on the results of the regression modeling, every one-unit increase in the natural-log transformed p22 titer was associated with a 0.26 percent decrease in total gray-matter volume or a 1.31 percent decrease in total gray-matter volume for a five-unit increase in the natural-log transformed p22 titer. Similarly, every one-unit increase in the natural-log transformed sag1 titer was associated with a 0.32 percent decrease in total gray-matter volume or a 1.6 percent decrease in total gray-matter volume for every five-unit increase in the natural-log transformed sag1 titer. While small, the differences in gray-matter volume associated with a five-unit increase in p22 or sag1 titers is consistent with the amount of volume loss associated with one year of aging [35]. Nonetheless, the clinical significance of the association between *Toxoplasma gondii* and gray-matter volume is unknown. Further, because of limitations associated with the cross-sectional design of our study, it is unknown whether the smaller gray-matter volume associated with *Toxoplasma gondii* is static or progressive.

An important consideration regarding our findings showing no associations between *Toxoplasma gondii* and gray-matter volume in the prefrontal cortex, hippocampus, and thalamus is that of sample size. To better interpret these negative findings, we conducted post-hoc power analyses to show the sample size needed to detect an effect on gray-matter volume from one year of aging in elderly subjects, reasoning that this would be a meaningful effect size from *Toxoplasma gondii*. The results of these power analyses showed that our study had 80 percent power to detect an effect size equal to the size that previous studies have associated with the effect of one year of aging in older adults [35, 36] for the left frontal pole, the left and right orbital frontal cortex, the left frontal operculum, the left hippocampus, and the left and right thalamus but not for the right frontal pole, the left and right superior frontal gyrus, the left and right orbital frontal cortex, the right frontal operculum, and the right hippocampus. As such, based on our sample size, several of the analyses were underpowered, and, accordingly, we do not know if in the underpowered analyses that there was no association or whether the lack of a significant association was due to lack of power from small sample sizes. In the analyses that were adequately powered, we are more confident concluding that there was no association.

In addition to the small sample size of our study, several other factors require consideration when interpreting our findings. One is that in the analyses of the prefrontal region, the hippocampus, and the thalamus, we examined only gray matter and do not know whether *Toxoplasma gondii* could be associated with white-matter volume. Similarly, we did not investigate gray matter in other individual regions. Gray-matter volume in other brain regions could be vulnerable to *Toxoplasma gondii*. An additional consideration is whether different strains of *Toxoplasma gondii* could affect gray-matter volume, as different strains might differ in virulence [37]. The sample we used was from the United Kingdom, whereas *Toxoplasma gondii* strains in other regions could have different effects on gray-matter volume, and inadequate data currently address this issue. Further, our study is cross sectional. In addition to not knowing whether the possible loss of total gray-matter volume associated with *Toxoplasma gondii* is progressive, we do not know when the initial infection from *Toxoplasma gondii* occurred. Infection during critical developmental periods could affect gray matter differently from infection during middle age or late middle age. The cross-sectional design also precludes definitive cause-and-effect determinations in the association between *Toxoplasma gondii* and total gray-matter volume. While we tested for several interactions, we did not test for interactions with other infectious diseases. It is feasible that interactions between *Toxoplasma gondii* and other infectious diseases could influence gray-matter volume. While we included several covariates in our models that potentially could confound the association between *Toxoplasma gondii* and

brain volume, other variables that we did not include potentially could confound the association between *Toxoplasma gondii* and gray-matter volume, resulting in the possibility of residual confounding.

In conclusion, in this study of community-dwelling middle-aged and late middle-aged adults in the United Kingdom, *Toxoplasma gondii* serointensity was associated with total gray-matter volume but not with total white-matter volume. There were no associations between *Toxoplasma gondii* seropositivity and serointensity and gray-matter volume in the prefrontal cortex, the hippocampus, or the thalamus, although several of these analyses were underpowered. Together, these findings suggest that *Toxoplasma gondii* in humans might be associated with smaller total gray-matter volume, although whether there is progressive loss of gray-matter volume is unknown. Additional research investigating associations between *Toxoplasma gondii* and brain volume including in other regions of the world is needed.

## Supporting information

**S1 File.**
(DOCX)

## Acknowledgments

This research has been conducted using the UK Biobank Resource under Application Number 41535. We also acknowledge the participants in the UK Biobank.

## Author Contributions

**Conceptualization:** Shawn D. Gale, Dawson W. Hedges.

**Data curation:** Lance D. Erickson, Dawson W. Hedges.

**Formal analysis:** Lance D. Erickson, Bruce L. Brown.

**Investigation:** Shawn D. Gale, Dawson W. Hedges.

**Methodology:** Lance D. Erickson, Bruce L. Brown.

**Writing – original draft:** Lance D. Erickson, Shawn D. Gale, Dawson W. Hedges.

**Writing – review & editing:** Lance D. Erickson, Bruce L. Brown, Shawn D. Gale, Dawson W. Hedges.

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
