## [Decision Letter · Decision Letter 0]

18 Nov 2020

PONE-D-20-30012

Association between Toxoplasma gondii Seropositivity and Serointensity and Brain Volume in Adults: A Cross-Sectional Study

PLOS ONE

Dear Dr. Gale,

Thank you for submitting your manuscript to PLOS ONE. After careful consideration, we feel that it has merit but does not fully meet PLOS ONE’s publication criteria as it currently stands. Therefore, we invite you to submit a revised version of the manuscript that addresses the points raised during the review process.

In addition to the comments from the Reviewers, I see that “gender” is used throughout the manuscript, although "sex" is also used. Usually, “sex” (the biological designation) is meant. “Gender” is the social construct and is rarely relevant in the context of the current manuscript. Please revise the text to use “sex” rather than “gender” throughout.

We look forward to receiving your revised manuscript.

Kind regards,

Niels Bergsland

Academic Editor

PLOS ONE

Journal Requirements:

Reviewers' comments:

Reviewer's Responses to Questions

**Comments to the Author**

1. Is the manuscript technically sound, and do the data support the conclusions?

Reviewer #1: Yes

Reviewer #2: Yes

2. Has the statistical analysis been performed appropriately and rigorously? 

Reviewer #1: Yes

Reviewer #2: Yes

3. Have the authors made all data underlying the findings in their manuscript fully available?

Reviewer #1: Yes

Reviewer #2: No

4. Is the manuscript presented in an intelligible fashion and written in standard English?

Reviewer #1: Yes

Reviewer #2: Yes

5. Review Comments to the Author

Reviewer #1: Associating Toxoplasma gondii to regional brain volume is an interesting research question. The authors used data available from UKBioBank to assess the relationship between serological data for exposure to Toxoplasma gondii and five prefrontal brain regions, the hippocampus and the thalamus.

This is a very well written manuscript.

The authors discuss "changes" due to Toxoplasma gondii however this is a cross-sectional study and the authors are only able to estimate "associations".

Brain volume is not directly associated with cognitive ability so this is unlikely to explain reductions in those that are seropositive.

Minor points:

Table 1 Income should be in £ not lb.

Reviewer #2: -Add the uniqueness of this study compared to other studies to discuss the same issue.

-Add more on the basis of this disease in the introduction

-Add shortly about routine MR imaging of brain in this disorder

-Add images of brain volume for patients and controls

Discus role of imaging using these refs

-Discuss the merits and limitations of the technique applied

6. PLOS authors have the option to publish the peer review history of their article (what does this mean?). If published, this will include your full peer review and any attached files.

Reviewer #1: **Yes: **Gordon D. Waiter

Reviewer #2: **Yes: **Ahmed Abdel Khalek Abdel Razek MD

---

## [Author Response · Author response to Decision Letter 0]

5 Jan 2021

Reviewer #1: 

Associating Toxoplasma gondii to regional brain volume is an interesting research question. The authors used data available from UK BioBank to assess the relationship between serological data for exposure to Toxoplasma gondii and five prefrontal brain regions, the hippocampus and the thalamus. This is a very well written manuscript.

The authors discuss "changes" due to Toxoplasma gondii however this is a cross-sectional

study and the authors are only able to estimate "associations".

We fully agree that we can only estimate associations in this study. We have now either eliminated references to “changes” and have used “associations” instead or have made it clear that we mean association. 

Brain volume is not directly associated with cognitive ability so this is unlikely to explain

reductions in those that are seropositive.

This is a really good point. There is evidence, though, that in healthy individuals, total brain volume does correlate with intelligence (see meta-analysis by Pietschnig et al., 2015), although the mechanism is unclear and the overall correlations while significant are small. In addition, there is evidence that brain volume in late life is associated with concurrent cognitive function (Royle NA, et al., 2013). We have added these references and clarified that we were looking for a potential mechanism by which Toxoplasma gondii could be associated with changes in brain function including cognition. As such, we now make an argument in the revised version of the manuscript that brain volume requires consideration in understanding how Toxoplasma gondii might affect brain and cognitive function. 

Table 1 Income should be in £ not lb.

We have made this change.

Reviewer #2: 

Add the uniqueness of this study compared to other studies to discuss the same issue.

We indicate in the introduction section that there are very few studies that have investigated the association between volumetric brain measurements and Toxoplasma gondii. The paper by Horacek et al. (2012) used a slightly different neuroimaging technique (VBM) than did the UK Biobank to compare controls with and without Toxoplasma gondii as well as those with schizophrenia who were either seropositive or seronegative. The sample size of the controls was only 56, of whom only 13 were seropositive for Toxoplasma gondii. Our use of a much larger sample size makes our study unique and less prone to error. 

Add more on the basis of this disease in the introduction 

We have added additional information about the biology and epidemiology of Toxoplasma gondii in the introduction section. 

Add shortly about routine MR imaging of brain in this disorder

Routine MR imaging is not used clinically in latent toxoplasmosis. Further, few studies have investigated MR imaging in toxoplasmosis, and we have included in our paper the available studies that have used MR imaging in studying brain volume associated with Toxoplasma gondii.

Add images of brain volume for patients and controls

In our analyses, we used preprocessed MRI data made available by the UK Biobank and did not analyze the original MRI images ourselves. Although we recognize the intellectual value of presenting characteristic Toxoplasma gondii positive and negative brain images, the simple response to this request is that cannot because we do not have access to the images themselves. Furthermore, because we did not use the MRI images at any stage of our analyses, we feel that it could misrepresent the MRI data we did use as the product of our own analysis of the MRI images if we included images in our report. Please note that Table 1 contains data about brain volumes in the study sample. 

Discus role of imaging using these refs

MRI imaging is not used clinically in latent toxoplasmosis, and few findings from studies investigating MR imaging in latent toxoplasmosis are available. Our study is one of the few available studies reporting abnormalities in brain imaging of latent toxoplasmosis. As such, there is not enough information about the role of imaging in latent toxoplasmosis. Based on the findings currently available, we believe that the best we can do is to write as we have that additional research is required to better understand the associations between Toxoplasma gondii and brain volume. 

Discuss the merits and limitations of the technique applied

As discussed in the discussion section and other sections of the manuscript, we acknowledge the following merits: utilization of a large dataset, standardized automated neuroimaging methods, statistical adjustment for variables known to be associated with brain volume (age, sex, education etc.), and validated cognitive measures were all strengths of this study. We described the limitations including decreased power in some analyses, the inability to know when a given participant initially acquired the infection, focusing on gray-matter volume in prefrontal regions when white-matter pathways could have been affected in prefrontal regions, the potential difference that different strains of Toxoplasma gondii may be related to, and the fact that our subjects were in the middle age to late age range. These were all limitations. In the revised manuscript, we have added material about the potential for residual confounding and also mention that the cross-sectional design precludes determination of causal relationships. While presentation of potential weaknesses of our study is not necessarily exhaustive, we fell that it identifies what we think are critical issues that readers should be aware of as they read the results of the paper.

---

## [Editor Report · Decision Letter 1]

12 Jan 2021

Association between Toxoplasma gondii Seropositivity and Serointensity and Brain Volume in Adults: A Cross-Sectional Study

PONE-D-20-30012R1

Dear Dr. Gale,

We’re pleased to inform you that your manuscript has been judged scientifically suitable for publication and will be formally accepted for publication once it meets all outstanding technical requirements.

Kind regards,

Niels Bergsland

Academic Editor

PLOS ONE
---

## [Editor Report · Acceptance letter]

14 Jan 2021

PONE-D-20-30012R1 

Association between *Toxoplasma gondii* Seropositivity and Serointensity and Brain Volume in Adults: A Cross-Sectional Study 

Dear Dr. Gale:

I'm pleased to inform you that your manuscript has been deemed suitable for publication in PLOS ONE. Congratulations! Your manuscript is now with our production department. 

Kind regards, 

on behalf of

Dr. Niels Bergsland 

Academic Editor

PLOS ONE